# Phenology and Monitoring of the Lesser Chestnut Weevil (*Curculio sayi*)

**DOI:** 10.3390/insects13080713

**Published:** 2022-08-08

**Authors:** Camila C. Filgueiras, Denis S. Willett

**Affiliations:** 1Department of Biology, University of North Carolina Asheville, Asheville, NC 28804, USA; 2North Carolina Institute for Climate Studies, North Carolina State University, Asheville, NC 28801, USA

**Keywords:** chestnut, weevil, lifecycle, *Curculio*, phenology, *Castanea*

## Abstract

**Simple Summary:**

The lesser chestnut weevil (*Curcilio sayi*) is an emergent pest of chestnuts in the United States. Knowledge of this weevils phenology and the ability to monitor its populations will help us understand how this pest is emerging with expanding chestnut production and provide the means to begin to mitigate its effects on chestnut production. We explored the seasonal emergence of *C. sayi* in upstate New York through the use of traps and soil microcosms to understand when weevils begin to emerge and when the population peaks. We found that pyramid traps are most effective for monitoring and that populations have one generation that tends to peak late in the season. We also found that generational cohorts may stagger their emergence and delay leaving the soil for more than one year.

**Abstract:**

With the introduction in recent years of high-yield blight-resistant chestnut varieties, the commercial chestnut industry in the United States is expanding. Accompanying this expansion is a resurgence in a primary pest of chestnut: *C. sayi*, the lesser chestnut weevil. This weevil damages the nut crop and infestations can surge from 0 to close to 100% in as little as two years. Understanding the dynamics of this pest has been challenging. Most work was conducted in the 1900s and only recently has this weevil garnered renewed interest. Recent work on *C. sayi* phenology has been completed in Missouri but conflicted with anecdotal reports from northern growers. From 2019 to 2020, we used a combination of trapping and microcosm studies to understand both *C. sayi* phenology and the means of monitoring this pest. *C. sayi* populations were univoltine and peaked in mid-October. Pyramid traps were the most effective at capturing adult *C. sayi*. *C. sayi* larvae, pupae, eclosed adults, and emerging adults were recovered from microcosm experiments. These results suggest that *C. sayi* emerges later in the northern US with the potential for a single generation to emerge over multiple subsequent years. Understanding *C. sayi* phenology along with the means of monitoring forms the basis for effective management and control in commercial chestnut orchards.

## 1. Introduction

*Curculio sayi* (Gyllenhal) is truly the lesser of two weevils. Both *C. sayi* (the lesser chestnut weevil) and *C. caryatrypes* (Boheman) are principal pests of chestnut production in the United States. Females of these species lay eggs in developing nuts and the larvae that emerge from these eggs can cause massive losses in nut production. While the economic and ecological consquences of chestnut weevil infestation are serious, not much is known about these weevils due to the interesting history of chestnut trees and chestnut production in the United States.

American chestnut trees (*Castanea dentata*) once dominated the forests of the Eastern United States. Up until the early 1900s, an estimated four billion trees, accounting for more than 50% of the total basal area in eastern forests, grew on 800,000 km^2^ [1]. A staple of early American life, chestnut trees provided tannin-rich, decay-resistant wood used in everything from furniture to housing to musical instruments [2,3]. The value of chestnut timber in Pennsylvania alone in 1912 was assessed at USD 55 million, equal to USD 1.4 billion today [3]. The introduction of chestnut blight (*Cryphonectria parasitica*) in the early 1900s devastated American chestnut forests [1,2,4]. Few trees survived and stump sprouts became infected well before maturation, surviving only as small, multi-stemmed shrubs. Chestnuts, and their associated insect pests, disappeared from the national consciousness.

Because of this historical dynamic, very little is known about chestnut weevils in North America. Of what research is available, most of it was published well before the year 2000 and the increasing impact of anthropogenic climate and land-use changes of the 21st century. Early work [5] recognized the prevalence and severity of these pests, documenting high levels of infestation reaching 100% in some cases, with large quantities being seized and destroyed due to ‘worms, worms excreta, worm-eaten chestnuts, and decayed chestnuts’ which ‘were therefore liable to seizure for confiscation’ [6]. Early work also documented some aspects of the phenology and lifecycle of these weevils including how they are able to lay eggs inside developing chestnuts and details as to their development [7].

While this work is historically important, it did not gain much attention until recent advances in chestnut breeding created a resurgence in the commercial chestnut industry. The recent development of high-yield chestnut varieties that are blight-resistant have increased the opportunities to expand production of this high-value specialty crop in the US [8,9,10,11,12]. New varieties of trees can produce 1000 to 1500 lbs per acre annually with prices ranging between USD 2.50 and USD 10 per pound depending on the market. As growers recognize the potential of commercial chestnuts, acreage in the northeast and Great Lakes regions is increasing and nurseries are rushing to keep up with demand for new trees (Zarnowski and Zarnowski, personal communication, 2020). Accompanying this resurgence is a resurgence in another organism: the lesser chestnut weevil.

As more blight-resistant trees come into production, weevil populations are exploding. If left unchecked, weevil populations can develop rapidly, reaching high levels of infestation in as little as two years [7,13,14]. Damage from this weevil is now recognized to be devastating in two forms. First, weevil larvae physically damage the nut and erode consumer confidence when emerging from purchased nuts. Second, larvae infestations are accompanied by fungal infections by *Aspergillus* fungi [15], which produce the diarrheagenic toxin emodin [16].

Efforts to control this pest depend on understanding its phenology. Investigations into *C. sayi* phenology recommenced in 2008 with work in Missouri that documented the emergence, adult activity, and lifecycle of the lesser chestnut weevil over three years [13]. Interestingly, the greater chestnut weevil was not encountered [13]. In 2019, chestnut growers in upstate New York began to have similar problems. Curiously, they commented, their phenologies did not match that reported in the literature from Missouri. In discussions with other colleagues and growers in the Northeast, two important anecdotal trends became readily apparent. First, the lesser chestnut weevils (not the greater) were rapidly emerging as a principal pest of chestnuts in the region, in some cases going from 0% to above 80% infestation in as little as two years. Second, literature and extension reports from southern states did not match experiences in the northeast. Growers would begin monitoring in early spring, not find any weevils, conclude there was no problem, and still find devastating infestation at the end of the season.

To begin to address these trends, we began working to better understand the phenology and monitoring of the lesser chestnut weevil as a first step to improving its management in the northeastern United States.

## 2. Materials and Methods

All *C. sayi* weevils were collected from Rose Valley Farm (43°09′26.5″ N, 76°55′21.1″ W), an organic farm in upstate Rose, New York during the field seasons of 2019 and 2020. These weevils were collected from a mature (15+ years old) commercial chestnut stand containing a mix of American Chestnut Hybrids. Trees were approximately 7 m on center and formed a complete canopy after leafing out. Catkins (flowering) occurred in mid to late June and nut drop began in late September to early October. The soil type was Elnora Loamy Fine Sand [17].

### 2.1. Phenology and Monitoring

To examine the phenology and evaluate the adult monitoring potentials of different trap types, three different traps were deployed to the mature chestnut stand at Rose Valley Farm at the beginning of the field season (May). Emergence traps were conical traps 1 m in diameter that were constructed similar to those described by (Keesey, 2008) from fine hardware cloth and placed directly over the soil under the mid-canopy of chestnut trees [13]. Trunk traps (Circle Trunk Trap, Small GL-4000-06, Great Lakes IPM Vestaburg, MI) were affixed to tree trunks at breast height (1.35 m above the ground) and consisted of fine mesh screen wrapped such that it abutted the diameter of the tree and terminated in a clear plastic trap. Pyramid traps (Tedders Pyramid Trap, GL-5000-06, Great Lakes IPM, Vestaburg, MI, USA) were staked into the ground within 0.5 m of the tree trunk and consisted of black upright cardboard supports terminating in a clear plastic trap.

A total of 16 pyramid traps, 12 trunk traps, and 3 emergence traps were emplaced in the chestnut stand. Traps were monitored weekly following emplacement and all insects in the trap were collected for evaluation in the lab. Male and female *C. sayi* totals for each trap were tallied on a biweekly basis in both 2019 and 2020. Specimens were examined for the presence of *C. caryatrypes* using diagnostic characteristics described previously [6,7,13,18].

Degree days were also monitored for each season. Degree days were calculated with a base of 4 °C using the Baskerville–Emin method beginning April 1. Temperature data were collected by the Network for Environment and Weather Applications (NEWA) station at Butler (Tree Crisp), NY, less than 17 km from the chestnut stand.

### 2.2. Lifecycle

To evaluate the lifecycle of *C. sayi*, 25 late stage *C. sayi* larvae that had recently emerged (within 24 h) from fallen chestnuts were placed into each of 20 soil microcosms adjacent to the commercial chestnut stand described above. These microcosms were constructed using 5-gallon buckets (one 5-gallon bucket for each of the 20 microcosms) perforated with 15–20 large-diameter (~7 cm) holes screened with fine mesh and filled with soil from the chestnut stand. These buckets were then covered with the same fine mesh and buried such that the soil level of the microcosm was flush with that of the adjacent soil. This set up allowed the transit of soil microorganisms through the fine mesh but prevented both the escape of the *C. sayi* larvae through the soil and the escape of any emerged adults above the surface.

These microcosms were emplaced in November of 2020. The 25 larvae were placed at the surface of the soil and the screen covering affixed to prevent escape. Microcosms were left unperturbed and then removed for analysis in late April 2021. To examine these microcosms, the top screen was first removed and the surface examined for adult *C. sayi*. Following the surface examination, the surface height of the soil was marked on the side of the bucket and the soil of the microcosm finely and delicately excavated and passed through a mesh sieve to look for larvae, adults, pupae, and pupal cells.

### 2.3. Analysis

Data were collated in tabular form (comma-separated values) and then imported to R V4.2.0 using RStudio as an IDE. The weevil catch was modeled using generalized linear models based on Poisson distributions using the log link function. All possible models were considered (including all levels of interactions) and best fit models were selected based on examination of diagnostic residual plots, likelihood ratio tests, analysis of deviance, information criteria, and goodness of fit. Differences in grouping factors were evaluated using estimated marginal means and Tukey’s pairwise contrasts adjusted for the family-wise error rate. The Tidyverse package was used to facilitate analysis and plotting [19]. The car, emmeans, and lmtest packages were use to assist in model development, analysis, and interpretation [20,21,22].

## 3. Results

### 3.1. Phenology

To understand *C. sayi* population dynamics and phenology, we evaluated trap catch on a biweekly basis during the growing season. Very few weevils, either male or female, were caught in traps until the beginning of September. *C. sayi* male and female populations spiked noticeably between September and November (Figure 1A). After the beginning of November, collection was discontinued following hard freezes. Only lesser chestnut weevil adults (*C. sayi*) were collected; no greater chestnut weevils (*C. caryatrypes*) were detected.

More male than female *C. sayi* were consistently captured during the growing season (Figure 1A). Male catch tended to rise earlier and faster than female trap catch. Both male and female *C. sayi* catch tended to peak mid-October.

The emergence of weevils like *C. sayi* is often associated with degree days. We evaluated cumulative degree days to associate population spikes with cumulative degree days. The collection of more than ten *C. sayi* weevils (blue vertical lines in Figure 1B) occurred between 2250 and 2500 cumulative degree days. Population spikes in mid-October were associated with approximately 2500 degree days.

### 3.2. Monitoring

We evaluated the efficacy of three different trap types for monitoring adult *C. sayi*. Pyramid and trunk traps were more effective for *C. sayi* monitoring (Figure 2A). Emergence traps (Figure 2B) did not catch any weevils in either field season. Sex, month, and trap type significantly explained observed weevil catch ($\chi^{2}$ = 99.4, 173.4, 304.7, df = 1, 1, 1, *p* < 0.0001, respectively). Pyramid traps (Figure 2D) caught 3.35 ± 0.34 (SE, z = 12.063, *p* < 0.0001) times more *C. sayi* adult weevils than trunk traps (Figure 2C). Male *C. sayi* adults were caught at rates 2.4 ± 0.227 (SE, z = 9.48, *p* < 0.0001) times that of *C. sayi* adult females.

### 3.3. Life Cycle

Adult female *C. sayi* lay their eggs in chestnut burrs (Figure 3A—mature chestnut burr), where the larvae burrow through the nut, consuming nut meat as they progress through instars. In late fall, late-instar chestnut weevils make a hole in the chestnut and emerge from the nut (Figure 3B). These larvae (Figure 3C) fall from the nut and burrow into the ground, where they construct a chamber from the soil in which to pupate. These pupae (Figure 3D) can remain in these chambers for some time and then eventually eclose into mature adults (Figure 3E).

Of the 500 *C. sayi* larvae originally introduced to the microcosms, only 1 larva, 1 pupa (Figure 4A,B), and 56 adults (Figure 4C,D) were recovered, none of which had emerged from the soil. The larva had constructed a pupal chamber but had not pupated. While a few adults had emerged from the pupal chamber, most were firmly ensconced in their protective cell and demonstrated no signs of active emergence.

Chestnut weevil pupae collected from pupal chambers were approximately 8 mm in length (Figure 4A,B). The pupal chambers in which they resided had internal chambers slightly larger than the pupae but outer diameters ranging from 2.7 cm to 3.5 cm. Eclosed adults (Figure 4C,D) were also found in pupal chambers. These adults were fully developed but had made no discernable effort to leave the chamber. Pupal chambers were only found between 5 and 10 cm below the soil line. Adult *C. sayi* were found both in pupal chambers and outside the chamber above 10 cm, seemingly in the process of emerging.

Adult *C. sayi* lesser chestnut weevils display striking sexual dimorphism. Male *C. sayi* adults (Figure 5A) are approximately 10 mm in length with comparatively shorter rostrums (Figure 5B). Female *C. sayi* adults (Figure 5C) are approximately 15 mm in length with comparatively longer rostrums. The rostrum is perhaps the most apparent feature of the chestnut weevil in the field, immediately noticeable in traps (Figure 6A) and with unperturbed individuals (Figure 6B).

## 4. Discussion

Observed *C. sayi* population dynamics suggest that the upstate New York population is univoltine with a peak in mid to late October (Figure 1A). This is in contrast to reports from Missouri that there are two distinctive waves of emergence [13]. This difference could be attributed to differences in season dynamics; *C. sayi* in Missouri begins emerging in late May [13] when the weather is still rather cold (nighttime temperatures around 4 °C) in upstate New York. Trapped *C. sayi* populations in upstate New York only began to rise when cumulative degree days began to exceed 2000 (Figure 1B).

In upstate New York, the adult *C. sayi* population dynamics were consistent across two field seasons with one large population spike towards the end of the season beginning in early September (Figure 1A). These populations tended to peak in mid to late October; by early November, populations were in decline. In both seasons, male *C. sayi* trap catch began to rise earlier than female *C. sayi* trap catch.

A second point of comparison between monitoring reports outside of the northeast is trap efficacy. Work in Missouri demonstrated the efficacy of emergence traps for collecting adult weevils as they emerge from the soil. In fact, these ground-based emergence traps collected many more *C. sayi* than and in advance of other trap types [13]. In contrast, our emergence traps in upstate New York did not catch a single *C. sayi* adult in either season. Much more effective were the trunk and pyramid traps. Pyramid traps caught more than three times more weevils than trunk traps and were the most effective means of monitoring adult *C. sayi* populations (Figure 2). These traps caught more than two times more male than female *C. sayi* adults. Both the pyramid and trunk traps work by arresting adults as they emerge from the soil and climb to the canopy. While this may be true if *C. sayi* adults emerge proximal to the host trees, this may not be the case at our field site. If adult weevils were emerging from the soil underneath the chestnut trees under observation, we would have expected to have collected at least a few given the trap placement. Additionally, the peak populations of adult weevils we collected came in mid to late October at a time when the nuts themselves were almost mature. It could be our *C. sayi* were flying into the canopy of our commercial orchard from surrounding areas.

This could be supported by the flight ability of *C. sayi* and the attractive nature of the chestnut canopy during flowering. While the average flight distances for male and female adult *C. sayi* are 247.1 m and 226.6 m, respectively, the maximum observed flights are much longer, with a female 2 hr flight of more than 3 km and a male flight exceeding 2.5 km [23]. If *C. sayi* were flying into the chestnuts evaluated in this study, they were likely attracted to the flowering catkins. Chestnut catkins produce a number of key volatile organic compounds [24] that are highly attractive to both male and female *C. sayi* [25].

The *C. sayi* lifecycle aligns closely with previous reports including the construction of a pupal chamber just below the soil surface. As in previous reports, no weevils or pupal chambers were found more than 10 cm below the soil surface. The mortality of our subterranean *C. sayi* was more than double that of previous reports which could be related to differences in soil types and the presence of other soil organisms in the microcosms. In line with previous reports, we recovered multiple life stages, including adults that had eclosed but not yet emerged.

These staggered life stages could point to a survival strategy that has been documented in the European cousin of *C. sayi*: the chestnut weevil *Curculio elephas*. *C. elephas* uses a staggered strategy of emergence to distribute a single generation to emerge as adults over multiple years [26,27]. This plasticity enhances long-term chances of survival so that an entire generation does not emerge in one potentially bad year [26,27]. A similar strategy could be occurring in *C. sayi*. Given the varied life stages we discovered after a single season and similar results from previous studies [13], we suspect that a single generation of larvae may emerge as adults over multiple years.

## 5. Conclusions

The phenology of *C. sayi* from upstate New York detailed here is distinct in some aspects from reports from other locations. Specifically the univoltine nature and late emergence of *C. sayi* suggest that *C. sayi* has different population dynamics in more northern climes. Additionally, the efficacy of the pyramid traps in contrast to other trap types points to effective monitoring practices for chestnut growers. Through better understanding the phenology and lifecycle of *C. sayi* along with a means to monitor populations, the improved management of this emergent pest can become a reality. Indeed, this work reported here can become the basis for the explorations of the effective means of the biological control of this lesser of the two weevils.

## Figures and Tables

**Figure 1 insects-13-00713-f001:**
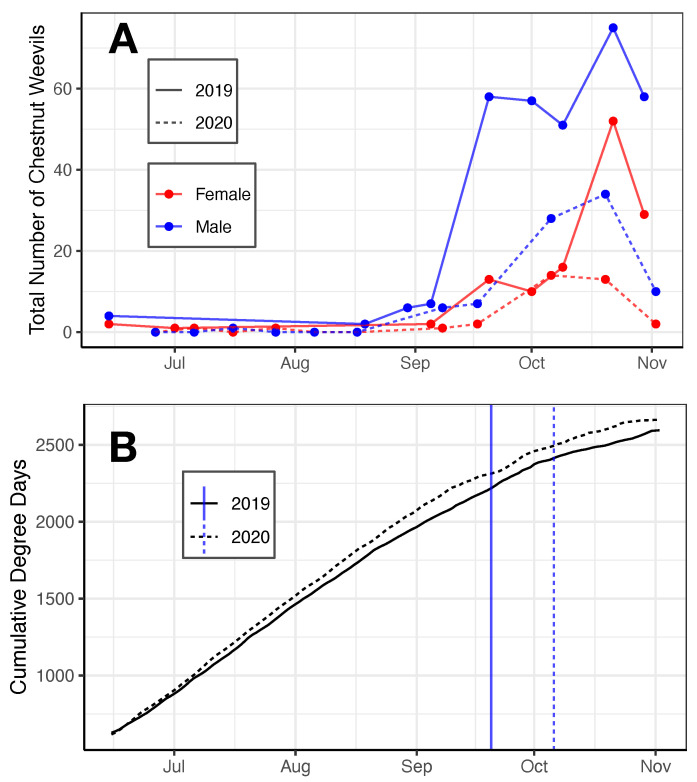
Phenology of the lesser chestnut weevil (*C. sayi*). (**A**) *C. sayi* male and female trap catch in 2019 and 2020. Data are summed across traps and trap types. (**B**) Degree day accumulations for 2019 and 2020. Degree days were calculated with a base of 4 °C using the Baskerville–Emin method beginning 1 April. Blue vertical lines denote first day in the season when more than 10 adult *C. sayi* individuals were collected.

**Figure 2 insects-13-00713-f002:**
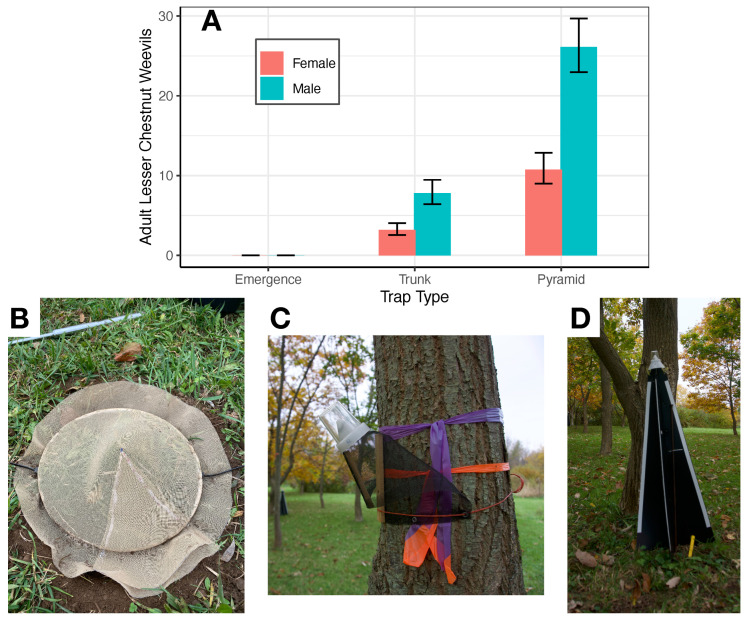
Monitoring the lesser chestnut weevil (*C. sayi*). (**A**) *C. sayi* male and female trap catch across three trap types. No weevils were ever caught in emergence traps. Bars and error bars denote average monthly catch and 95% confidence intervals, respectively. (**B**) Emergence trap/microcosm cover. This picture shows a conical cover for a buried microcosm. The emergence trap is functionally similar but 1 m in diameter. These traps were used to collect adult weevils emerging from the soil. (**C**) Trunk trap for collecting adult *C. sayi* moving upward along tree trunks. (**D**) Pyramid (Tedders) trap for collecting adult *C. sayi* emerging from the ground and moving toward dark upright objects.

**Figure 3 insects-13-00713-f003:**
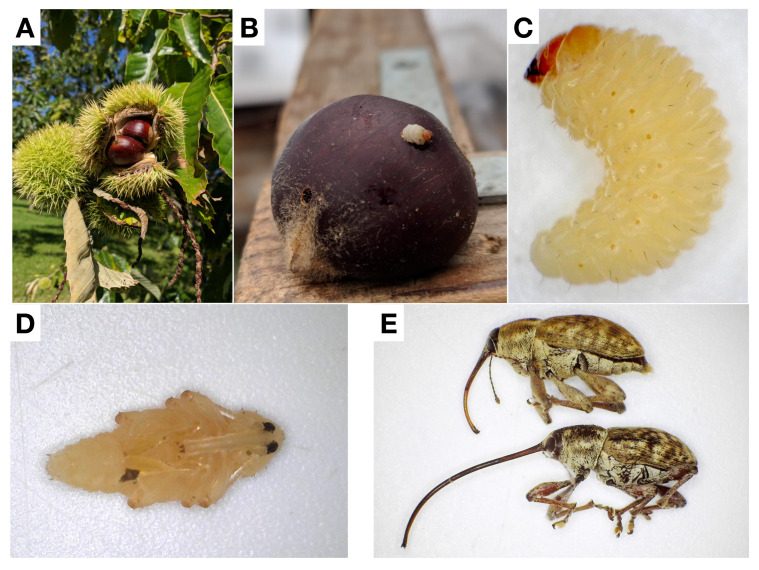
The lesser chestnut weevil (*C. sayi*) lifecycle. (**A**) *C. sayi* adult females lay eggs in chestnuts (depicted in their burrs). (**B**) *C. sayi* larvae feed inside the nut and, when they reach their last instar, create a hole and emerge from the chestnut. (**C**) *C. sayi* larvae emerge out of the holes and fall to the ground, where they burrow to about 5 cm below the soil line. (**D**) *C. sayi* pupate in the soil (extracted pupae depicted). (**E**) *C. sayi* pupae then eclose. Males (above with shorter rostrum) and females (below with longer rostrum) then emerge from the soil to continue the lifecycle.

**Figure 4 insects-13-00713-f004:**
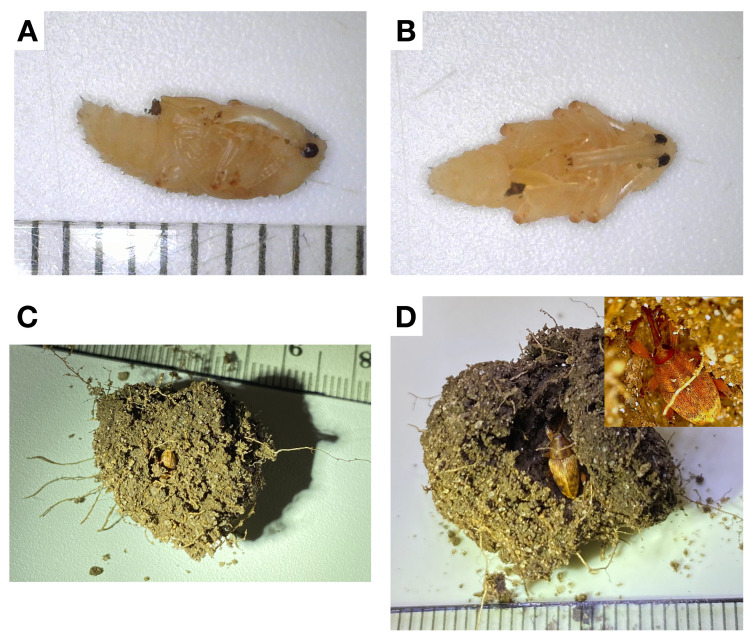
Subterranean life stages of the lesser chestnut weevil (*C. sayi*). (**A**) Side profile of *C. sayi* pupa. Larvae burrow into the soil to construct pupal chambers in which they pupate. Ruled divisions denote millimeters. (**B**) Ventral view of *C. sayi* pupa. (**C**) *C. sayi* adults eclose from pupae but can remain in the soil for a period of time in the pupal chamber. Small-ruled divisions denote millimeters and large-ruled divisions denote centimeters. The adult *C. sayi* weevil is oriented downward; only the posterior is visible. (**D**) *C. sayi* adult gently excavated from pupal chamber. Inset provides close up of head and rostrum. Ruled divisions denote millimeters.

**Figure 5 insects-13-00713-f005:**
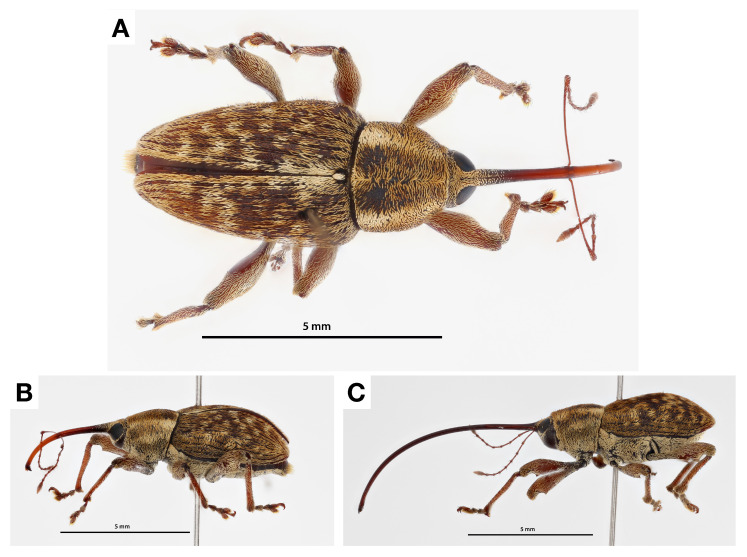
Adult lesser chestnut weevils (*C. sayi*). (**A**) Dorsal view of adult *C. sayi* male. (**B**) Side view of adult *C. sayi* male. (**C**) Side view of adult *C. sayi* female. Note difference in rostrum sizes.

**Figure 6 insects-13-00713-f006:**
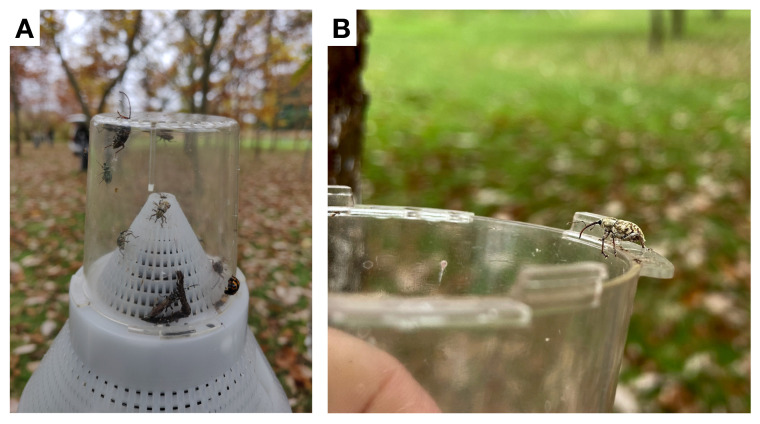
Adult lesser chestnut weevils (*C. sayi*) were collected in trunk and pyramid traps using a conical plastic mesh that terminated in a clear plastic cup perforated at the end. (**A**) Many adult *C. sayi* were eager to escape. Note rostrum of adult female *C. sayi* extending above the trap. (**B**) Some *C. sayi* adults preferred to hang around.

## Data Availability

Data and code supporting this publication will be made available via GitHub following publication.

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
