# Peer review of "Phenology and Monitoring of the Lesser Chestnut Weevil (Curculio sayi)"

_insects, 2022, doi:10.3390/insects13080713_

Round 1
Reviewer 1 Report
According to the authors, Lesser Chestnut Weevil (Curculio sayi) is resurging as a primary pest of chestnut (Castanea dentata) with the expansion of commercial high-yield blight resistant chestnut varieties in the United States. However, reports about this pest phenology are distinct between distinct locations in the United States. In order to clarify and to better understand C. sayi population dynamics and phenology, and monitoring traps; authors carried out trapping and microcosm studies during two years, testing the efficacy of three different trap types for monitoring, on a biweekly check during the growing season. The authors believe that the results of this study will contribute to improve its management in the northeastern United States.
In my opinion the Ms is well written, methodologies and results are very clear. However, I have some concerns about novelty and scientific interest, because seems to me very specific and restrict; and in my opinion, this Ms would fit better as a technical report than a scientific article.
Few edits:
1. Introduction
Page 2. …in anther organism… (another?)
2. Material and Methods
2.1. Phenology & Monitoring
Page 2. …and evaluate the monitoring potential… (maybe add adult monitoring)
Page 3. …(Figure 2B). Trunk traps… (Figure 2 appear first than figure 1)
Page 3. … Degree Days were calculated with a base of 4oC using the Baskerville-Emin method beginning April 1. (maybe you could describe it immediately after you mentioned it at the beginning of the paragraph!)
Page 3. … each of 20 soil microcosms adjacent… (20 soil microscosms, differents? Maybe it would be good if more information about soil microscosm is given!
Soil used was sterilized? and analyzed?)
…using 5 gallon buckets perforated… (5 gallon for all microcosms? for me is not totally clear?)
…The 25 larvae were placed at the surface of the soil and the screen covering affixed to prevent escape…. (I think it should be mentioned before!)
2.2. Analysis (Maybe in this section you can mention the explanatory variables used in the analysis...and interactions, if contemplated!?)
Results
Page 4. …occurred between 2000 and 2500 cumulative degree days…. (more than ten, I would adjust to 2250 and 2500 cumulative degree days!?)
3.2. Monitoring
…types for monitoring C. sayi…. (maybe add adult?)
3. Discussion (I would suggest to remove Figure citations from Discussion section)
Page 9. ... These traps caught more than two times more male than female C. sayi adults… (Maybe this should be in results too)
Author Response
We really appreciate your detailed comments and feedback. Please see the attached letter detailing our incorporation of your suggestions.

Reviewer 2 Report
1. Write down the purpose and objectives of the research well.
2. The figures in the manuscript are arranged incorrectly. Figures should be placed after their first mention in the text.
3. What do the authors suggest to combat the C. sayi?
Author Response
Thank you for your reviews, feedback, and suggestions. Please see the attached document outlining our revisions.
